# Disentangling Regional Primitives for Image Generation

## Abstract

This paper presents a method to explain the internal representation structure of a neural network for image generation. Specifically, our method disentangles primitive feature components from the intermediate-layer feature of the neural network, which ensures that each feature component is exclusively used to generate a specific set of image regions. In this way, the generation of the entire image can be considered as the superposition of different pre-encoded primitive regional patterns, each being generated by a feature component. We find that the feature component can be represented as an OR relationship between the demands for generating different image regions, which is encoded by the neural network. Therefore, we extend the Harsanyi interaction to represent such an OR interaction to disentangle the feature component. Experiments show a clear correspondence between each feature component and the generation of specific image regions.

## 1 Introduction

The interpretability of deep neural networks (DNNs) has received increasing attention along with the fast development of deep learning. However, there is a clear technical boundary between techniques of explaining a single scalar output score of a DNN[1] and methods of explaining the high-dimensional output (e.g., an image) of a DNN. For example, for DNNs for classification, attribution methods (Simonyan, 2013; Shrikumar et al., 2017; Selvaraju et al., 2017) were developed to estimate attributions of input variables to the scalar classification confidence. Zeiler & Fergus (2014) and desai & Ramaswamy (2020) visualized inference patterns encoded by a DNN, which determined the scalar classification confidence. In contrast, for image generation, the generated image contains much richer information than a scalar classification confidence, so it is difficult to apply previous explanation techniques designed for a single scalar output of a DNN. Instead, image generation is usually explained by controlling image generation results via input engineering (Härkönen et al., 2020; Voynov & Babenko, 2020).

Therefore, the essence of such difference in the explanation techniques lies in the two facts. (1) The explanation of a single scalar output[2] can be considered to explain the *the structure of a DNN's inference logic.* (2) In comparison, there is *no solid theory developed to explain* the internal representation structure of the neural network for image generation. For example, as Figure 1 shows, both attribution/importance scores of input variables to the classification confidence (Selvaraju et al., 2017) and interactions between input variables for inference (Ren et al., 2024a; 2023a) all reflect potential representation structure of the DNN.

However, how to explain the internal representation structure of an image-generation model has not been sophisticatedly formulated. This problem can be discussed in the following two aspects.

First, we can empirically say that most DNNs do not encode image information at a pixel level[3]. Instead, image generation of the DNN is conducted somewhat like pasting a set of pre-encoded regional patterns, rather than let each pixel be generated independently. **Therefore, how to faithfully**

---

[1]Outputs of these models are supposed to be summarized as a single or very few scalar scores.

[2]For example, people usually explain the scalar classification confidence $\log \frac{p(y=y^{\text{truth}}|\mathbf{x})}{1-p(y=y^{\text{truth}}|\mathbf{x})}$ of most discriminative models for multi-category classification

[3]PixelRNN and PixelCNN (Van Den Oord et al., 2016) are special cases, but these models still encode spatial relationships between pixels into larger patterns, instead of handling each individual pixel independently.

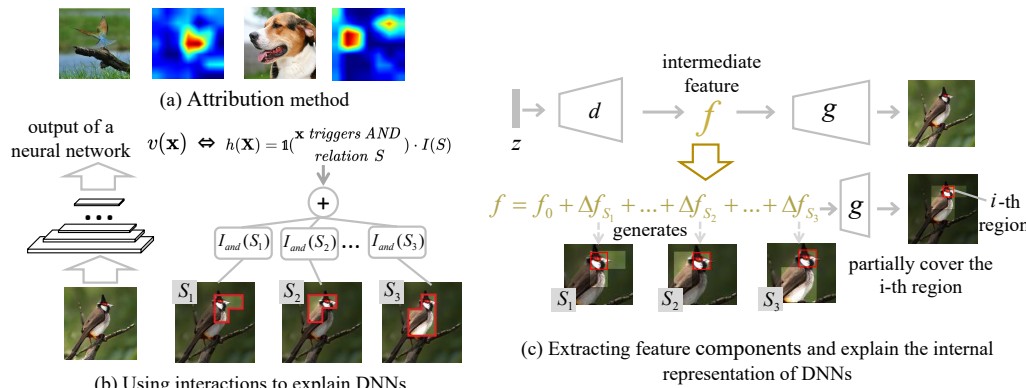

Figure 1: Different ways of explaining representation structures of a DNN. Both estimating the attribution of input variables (a) and extracting interactions between input variables encoded by the DNN (b) partially explain the representation structure of a DNN for image classification. (c) In comparison, we propose to decompose the feature in an intermediate layer $f$ into different feature components $\Delta f_1, \Delta f_2, ..., \Delta f_m$. Each $i$-th feature component $\Delta f_i$ is exclusively used to generate an primitive regional pattern in $S_i$. Thus, the generation of an entire image can be explained as the superposition of all primitive regional patterns.

**formulate and quantify the primitive regional patterns is the core of explaining the internal representation structure of image-generation models.**

Second, is it possible for the image generation result to be mathematically represented as the superposition of different primitive regional patterns?

Therefore, in order to formulate the representation structure of a given DNN for image generation, in this study, we disentangle the feature $f$ of an intermediate layer of the DNN into different feature components $f = f_0 + \Delta f_1 + \Delta f_2 + ... + \Delta f_m$, each being used to generate a specific subset of image regions. Specifically, as Figure 1 shows, the disentanglement of feature components should satisfy the following two requirements.

• Each feature component $\Delta f_i$ is responsible for generating a specific primitive regional pattern $S_i$.

• The generation of the entire image can be explained as the superposition of different primitive regional patterns.

In this way, above two requirements ensure the faithfulness of the explanation of the image generation. The generation of a certain subset of image regions is exclusively determined as the superposition of different primitive regional patterns, each being generated by a specific feature component.

To this end, we prove that the feature component $\Delta f_i$ represents a certain OR relationship between the demands for the generation of different image regions. We extend the theory of the Harsanyi interaction (Harsanyi, 1958) to mathematically formulate the OR relationship between different primitive regional patterns used by a given neural network. Our theory ensures that the generated image can be represented as a linear superposition of these primitive regional patterns.

As Figure 2 shows, the OR interaction means that when the DNN is required to generate any one region of $S_i$, $\Delta f_i$ must be added into the feature in the intermediate layer.

In sum, this paper proposes a new method to disentangle feature components from the intermediate layer of a DNN. It is theoretically guaranteed that each feature component is exclusively used to generate a specific set of image regions. Experiments have demonstrated the faithfulness of the proposed explanation of image generation.

## 2 RELATED WORK

**The interpretability analysis of GAN model.** In recent years, numerous studies have explored the interpretability of GAN models. Some studies explained the GAN models by analyzing the latent

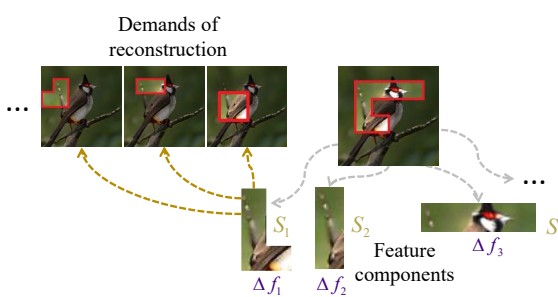

Figure 2: The selection of feature components reflects an OR relationship between demands of reconstructing different regions. Given the demand of reconstructing a certain set of image regions, a feature component $\Delta f_k$ is selected if its action region $S_k$ contains any one target image region for reconstruction.

space of GANs (Abdal et al., 2021; Härkönen et al., 2020; Lang et al., 2021; Patashnik et al., 2021). For instance, Härkönen et al. (2020) employed principal component analysis (PCA) to identify the main directions in the latent space. By making layer-wise perturbations along these directions, it achieved interpretable control of the image generation process. While most of these works focused on GAN interpretability from the perspective of input manipulation, some other works took a different approach by examining the role of intermediate neurons in the generation process. For example, Bau et al. (2018) identified specific neurons associated with certain object categories. In this way, this method enabled people to control the presence of certain objects in the generated images by adjusting the activation of the neurons.

Unlike previous output control based on input engineering, we aim to explore the internal representation structure of an image-generation network, which represents a new explanation perspective. We disentangle feature components that control the generation of specific regions, and we discover that the feature component can be formulated as the OR relationship between the demands of reconstructing different image regions.

**Interaction-based DNN explanation**. In the field of explainable AI, an emerging question is whether the decision-making process of DNNs can be interpreted as a set of sparse symbolic concepts. To address this, a theoretical system based on the Harsanyi interaction has been proposed to explain how symbolic concepts are encoded by DNNs. Over the past three years, about 20 articles have been published in the field of explainable AI, which aimed to tackle the mathematical possibility of explaining DNN inference logic through a limited set of logical patterns. Most of these studies were surveyed by Ren et al. (2024a). Specifically, Sundararajan et al. (2020); Janizek et al. (2021); Tsai et al. (2023) proposed different types of interactions between input variables of a DNN. Ren et al. (2023a) used the Harsanyi dividend (Harsanyi, 1958) to represent the AND interaction in a DNN. They also experimentally discovered that DNNs usually encoded a limited set of interactions between input variables, *i.e.,* the sparsity of AND interactions. Li & Zhang (2023) revealed that low-order interactions exhibited higher transferability across different input samples in discriminative neural networks. Ren et al. (2024a) proved the three common conditions under which the sparsity of interactions could be guaranteed. Ren et al. (2023b) proposed a method to learn optimal masked states of input variables based on interactions and alleviated the bias of the Shapley value caused by the sub-optimal masked states of input variables. Furthermore, Chen et al. (2024) extracted common interactions across different neural networks, and interactions shared by different neural networks usually represented generalizable inference patterns. Cheng et al. (2024) proposed to extract the interactions from the intermediate layers of neural networks, so as to illustrate how DNNs gradually learned and forgot inference patterns during forward propagation.

In addition, the interaction theory can also explain the representation power of a neural network. Specifically, this can be elaborated as follows. Wang et al. (2020) discovered and proved the negative correlation between DNNs' adversarial transferability and the interaction inside adversarial perturbations. Ren et al. (2021) found adversarial attacks primarily affected the high-order interactions than low-order interactions. Similarly, Zhou et al. (2024) revealed that low-order interactions tended to generalize better than high-order interactions. Liu et al. (2023) explained the intuition that DNNs learned low-order interactions more easily than high-order interactions. Deng et al. (2022) proved a counter-intuitive bottleneck *i.e.*, bivariate interactions of middle orders were usually not well encoded by a neural network. Ren et al. (2023c) found that Bayesian neural networks (BNNs) tended to avoid encoding complex Harsanyi interactions, compared to normal neural networks. Ren et al. (2024b) and Zhang et al. (2024) discovered and proved the dynamics of learning interactions

in neural networks exhibited a two-phase phenomenon, which had been widely observed in different neural networks and tasks. Deng et al. (2024) proved that the mechanisms underlying fourteen different classical attribution methods could all be rewritten as different redistributions of interaction effects to input variables.

In sum, most previous studies used interactions to explain the scalar output of a neural network. In comparison, in this paper, we first attempt to apply interactions to explain high-dimensional image generation of a DNN, which proposes fully new challenges. To this end, we discover that each feature component disentangled from the neural network can be formulated as an OR interaction between demands of reconstructing different image regions. Experiments verified the effectiveness of the proposed method.

## 3 EXPLAINING REPRESENTATION STRUCTURE OF IMAGE GENERATION

### 3.1 PRELIMINARY: AND INTERACTION

As the theoretical foundation of the explanation of an image-generation network, let us first introduce the definition of interactions in the task of image classification. Given an input image $\mathbf{x}$, let $v(\mathbf{x})$ denote the output of a DNN for image classification. We can set $v(\mathbf{x}) = \log \frac{p(y=y^{\text{truth}}|\mathbf{x})}{1-p(y=y^{\text{truth}}|\mathbf{x})}$ to denote the classification confidence. Let us divide the image $\mathbf{x}$ into $n = H \cdot W$ regions, and therefore we rewrite $\mathbf{x} = [\mathbf{x}_1, \mathbf{x}_2, \ldots, \mathbf{x}_n]^T$, where $\mathbf{x}_k$ denotes the $k$-th image region. We use $N = \{1, 2 \ldots, n\}$ as the set of indices of all image regions. For each specific set $S \subseteq N$ of image regions, the numerical effect of the Harsanyi interaction between image regions in $S$ is computed as

$$I(S) \overset{\text{def}}{=} \sum_{T \subseteq S} (-1)^{|S|-|T|} \cdot u(T) \tag{1}$$

where $u(T) \overset{\text{def}}{=} v(\mathbf{x}_T) - v(\mathbf{x}_\emptyset)$, and $v(\mathbf{x}_T)$ denotes the classification confidence *w.r.t.* the true category of the masked sample $\mathbf{x}_T$, where regions of $N \backslash T$ are masked, while regions in $T$ remain unchanged. Therefore, $u(N) = v(\mathbf{x}) - v(\mathbf{x}_\emptyset)$ represents the overall effect of all the input variables.

Each Harsanyi interaction represents an AND relationship between input variables encoded by the DNN, and contributes a certain effect $I(S)$ to the network output $v(\mathbf{x})$. For example, as shown in Fig 1(b), the DNN encodes the non-linear relationship between $S = \{head, mantle, body...\}$ to form a bird pattern. The co-appearance of all images regions in $S$ triggers the AND relationship of the bird pattern and makes an effect $I(S|\mathbf{x})$ to the classification confidence $u(N)$. The absence of any region in $S$ will remove the effect $I(S|\mathbf{x})$ from $u(N)$.

**Ren et al. (2024a) and Zhou et al. (2023) have discovered and partially proven the sparsity property and the universal-matching property of the interactions**, as the mathematical guarantee for taking interactions as faithful primitive inference patterns encoded by the DNN. According to Theorem 3.1, we can construct a surrogate logical model $h(\cdot)$ based-on the extracted interactions. This surrogate logical model can accurately fit the classification confidence $u(\cdot)$ of a DNN, no matter how the input is masked, *i.e.*, $\forall T \subseteq N, u(T) = h(\mathbf{x}_T)$. The above property is called *the universal-matching property*.

**Theorem 3.1.** *(Universal-matching property (Ren et al., 2024a), also proved in Appendix B). Given an input sample $\mathbf{x}$, the output $u(T)$ on each masked sample $\{\mathbf{x}_T \mid T \subseteq N\}$ can be well matched by a surrogate logical model $h(\mathbf{x}_T)$. The surrogate logical model sums up effects of all interactions that are triggered by the masked sample $\mathbf{x}_T$ as the output score.*

$$\forall T \subseteq N, u(T) = h(\mathbf{x}_T),$$

$$subject\ to\ h(\mathbf{x}_T) \overset{def}{=} \sum_{S \subseteq N, S \neq \emptyset} \mathbb{1}\left(\begin{array}{c} \mathbf{x}_T\ triggers\ AND \\ relation\ S \end{array}\right) \cdot I(S) = \sum_{S \subseteq T, S \neq \emptyset} I(S) \tag{2}$$

Besides *the universal-matching property*, the *sparsity property* is another property that guarantees the faithfulness of the interaction-based explanation. When we enumerate all subsets $S \subseteq N$ and compute effects of all $2^n$ interactions, it is proved that for the DNNs with stable outputs on masked input samples,[4] most of the extracted interactions have negligible effects on the output, *i.e.*,

---

[4] Ren et al. (2024a) proposed the common conditions for smooth outputs on masked inputs, which could be satisfied by most well-trained DNNs.

$I(S|\mathbf{x}) \approx 0$. Only a small set of interactions, denoted by $\Lambda = \{S \subseteq N : |I(S)| > \tau\}$, have considerable effects , where $\tau$ is a small scalar threshold. Therefore, we consider such a small number of salient interactions as the faithful explanation of the DNN. As Corollary 3.2 shows, the surrogate logical model $h(\cdot)$ can be approximated by these salient interactions.

**Corollary 3.2** (*Sparsity property (Ren et al., 2024a)*). *The surrogate logical model $h(\mathbf{x}_T)$ on each randomly masked sample $\mathbf{x}_T$, $T \subseteq \mathcal{N}$, uses the sum of a small number of salient interactions to approximate the network output score $u(T)$.*

$$\forall T \subseteq N, \qquad u(T) = h(\mathbf{x}_T) \approx \sum_{S \subseteq T, S \neq \emptyset} I(S) \tag{3}$$

### 3.2 TWO REQUIREMENTS TO EXPLAIN REPRESENTATION STRUCTURE

In this paper, we aim to extract the internal representation structure encoded by a DNN for image generation. Instead of generating each pixel independently[5], lots of empirical findings (Härkönen et al., 2020; Voynov & Babenko, 2020) all showed that a DNN usually encoded a set of regional patterns as the internal structure of an image, and the generation of an image could be considered as the superposition of the pre-encoded regional patterns.

Therefore, in this paper, we extend the Harsanyi interaction to represent such regional patterns from a trained DNN. Let a DNN generate an image $\mathbf{x}$ with a certain input code. We divide the generated image into $n = H \cdot W$ images, denoted by $\mathbf{x} = [\mathbf{x}_1, \ldots, \mathbf{x}_n]^T$. $N = \{1, 2 \ldots, n\}$ denotes the set of indices for image regions. Let $f \in R^D$ denote the feature of an intermediate layer of the DNN. Then, *the objective of this study is to decompose the feature $f$ into $m$ feature components, so as to let each feature component $\Delta f_k$ exclusively generate a specific regional pattern.*

$$\mathbf{x} = g(f), \qquad f = f_0 + \Delta f_1 + \Delta f_2 + \ldots + \Delta f_m \tag{4}$$

where $f_0$ represents a baseline feature as a non-informative feature state, and $M = \{1, 2, ..., m\}$ denotes the index set of all feature components. $f_0$ can be set as the average feature over all features given different input codes $\mathbf{z}$. *i.e.*, $f_0 = \mathbb{E}_{\mathbf{z}}[d(\mathbf{z})]$, where $d(\cdot)$ denotes the modules of the image-generation model between the input code $\mathbf{z}$ and the intermediate feature. We use $g(f)$ to represent the image generated by the feature $f$, *i.e.*, $\mathbf{z} \xrightarrow{d} f \xrightarrow{g} \mathbf{x}$.

**Two requirements for feature decomposition.** In this way, each feature component $\Delta f_k$ added upon the baseline feature $f_0$ is supposed to exclusively generate a certain regional pattern $S_k \subseteq N$, and thus can be taken as the primitive patterns encoded by the DNN. $S_k$ is also termed the ***action region*** of the $k$-th feature component. In this way, the generation of specific image regions is controlled by a set $\Omega \subseteq M$ of feature components, denoted by $F(\Omega) \overset{\text{def}}{=} f_0 + \sum_{k \in \Omega} \Delta f_k$. In particular, $F(M) = f$. To achieve this, the feature decomposition is conducted *w.r.t.* the following two requirements.

**Requirement 1:** *Each feature component $\Delta f_k$ exclusively generates a specific set of image regions $S_k \subseteq N$, without affecting other image regions. I.e., the addition of $\Delta f_k$ to $F(\Omega)$ should not change the generation of other regions.*

$$\forall k \in M, \quad \forall \Omega \subseteq M \setminus \{k\}, \quad g_{N \setminus S_k}(F(\Omega)) = g_{N \setminus S_k}(F(\Omega \cup \{k\}))$$
$$s.t. \quad F(\Omega \cup \{k\}) = F(\Omega) + \Delta f_k \tag{5}$$

$g_{N \setminus S_k}(\cdot)$ *denotes image regions in $N \setminus S_k$ selected from the generated image $g(\cdot)$.*

**Requirement 2-$\alpha$.** *The generation of each $i$-th image region can be fully determined by the superposition of all feature components that cover the $i$-th region.*

$$\mathbf{x}_i = g_{S=\{i\}}(F(\Omega)), \quad subject\ to \quad \Omega = \{k \mid i \in S_k\} \tag{6}$$

*i.e., $g(F(\Omega))$ well generates the $i$-th region $\mathbf{x}_i$ of the target image $\mathbf{x}$.*

**Requirement 2-$\beta$.** *The generation of image regions in $\hat{S}$ can also be exclusively determined by the superposition of feature components that intersect with $\hat{S}$, i.e.,*

$$\mathbf{x}_{\hat{S}} = g_{\hat{S}}(F(\hat{\Omega})), \quad subject\ to \quad \hat{\Omega} = \{k \mid S_k \cap \hat{S} \neq \emptyset\} \tag{7}$$

---

[5]Except for neural networks like PixelCNN and PixelRNN.

*where $\hat{\Omega}$ denotes the set of feature components whose action regions partially cover regions in $\hat{S}$.*

Requirement 1 shows that all regions in $S_k$ can be considered as a singleton visual pattern encoded by the DNN. The above requirements ensures that the feature decomposition is a faithful explanation of the representation structure of the DNN generating the image **x**.

**Minimal feature for regional generation.** We can consider that $\Delta f_1 + \Delta f_2 + ... + \Delta f_m$ are features required to be added to the baseline feature $f_0$ to generate the entire image **x**, in Equation (4). If we are only requested to reconstruct a subset $\hat{S}$ of image regions, then we only need to use feature components $F(\hat{\Omega}) = f_0 + \sum_{k \in \hat{\Omega}} \Delta f_k$, *s.t.* $\hat{\Omega} = \{k \mid S_k \cap \hat{S} \neq \emptyset\} \subseteq M$, and the addition of any further feature components will not affect the generation of regions in $\hat{S}$, according to above requirements. $\hat{\Omega}_{\hat{S}}$ denotes the set of feature components selected to reconstruct image regions in $\hat{S}$. In this way, we can consider $F(\hat{\Omega})$ as the ***minimal feature*** of generating image regions in $\hat{S}$, denoted by $u(\hat{S}) = F(\hat{\Omega})$. On the other hand, the minimal feature $u(\hat{S})$ can be also estimated as follows.

$$u(\hat{S}) = \mathrm{argmin}_{u(\hat{S})} \|u(\hat{S})\|_{\text{L-1}}, \quad w.r.t. \quad \mathbf{x}_{\hat{S}} = g_{\hat{S}}(u(\hat{S})). \tag{8}$$

Therefore, the decomposition of feature components $\Delta f_k$ *w.r.t.* above requirements can be written as follows.

$$\begin{aligned} &\textit{Decomposion of } f = f_0 + \sum_{k \in M} \Delta f_k \textit{ w.r.t. Requirement 1,2-}\alpha\textit{,2-}\beta \\ &\equiv \quad \min_{\{\Delta f_k\}} \|u(\hat{S})\|_{\text{L-1}} \textit{ s.t. } \forall \hat{S} \subseteq N, u(\hat{S}) = F(\hat{\Omega}) = f_0 + \sum_{k \in \hat{\Omega}} \Delta f_k \end{aligned} \tag{9}$$

where $\hat{\Omega} = \{k \mid S_k \cap \hat{S} \neq \emptyset\}$

**Implementation details.** The computation of the minimal feature $u(\hat{S})$ *w.r.t.* image regions in $\hat{S}$ can be approximated as $\min_{\alpha} \|u(\hat{S})\|_{L-1} + \frac{\lambda}{|\hat{S}|} \|g_{\hat{S}}(u(\hat{S})) - \mathbf{x}_{\hat{S}}\|_{L-2}^2$, *subject to* $\|g_{\hat{S}}(u(\hat{S})) - \mathbf{x}_{\hat{S}}\|_{L-2} < \tau$, $u(\hat{S}) = f_0 + \alpha \odot \Delta f$. Here, we consider the minimal feature are all *contained* by the whole feature $f$, so we apply the empirical constrain $u(\hat{S}) = f_0 + \alpha \odot (f - f_0)$, *w.r.t.* $\alpha \in [0,1]^D$. Each feature dimension of $\alpha$ is in the range of [0,1]. $\odot$ is referred to as the element-wise multiplication. In this way, we constrain $u(\hat{S})$ strictly locates within the range between $f_0$ and $f$.

### 3.3 USING INTERACTIONS FOR FEATURE DISENTANGLEMENT

The above subsection introduces how to compute the minimal feature component, and clarifies the mathematical connection between the feature components $\{\Delta f_k\}_k$ and minimal features $\{u(\hat{S})\}_{\hat{S}}$. Then, in this subsection, we introduce how to disentangle feature components $\Delta f_1, ..., \Delta f_m$ from all minimal features, so as to let each feature component $\Delta f_k$ exclusively generate regions in $S_k$.

**OR interaction.** To this end, we prove that the proposed two requirements can be rewritten as follows.

• For each feature component $\Delta f_l$ whose action regions does not cover any regions in $S_k$, *i.e.,* $S_k \cap S_l = \emptyset$, this component does not affect the generation of the image regions in $S_k$.

• $u(\hat{S}) = F(\hat{\Omega}) = f_0 + \sum_{k \in \hat{\Omega}} \Delta f_k, \quad \hat{\Omega} = \{k \mid \hat{S} \cap S_k \neq \emptyset\}$

Clearly, above two terms reflects an OR relationship in the selection of feature components towards the reconstruction of the target regions in $\hat{S}$. When we need to reconstruct image regions in $\hat{S}$, then the feature component $\Delta f_k$ should be added to $f_0$ if and only if $\hat{S}$ covers any regions in $S_k$. For example, as Figure 2 shows, let us consider a feature component $\Delta f_k$ *w.r.t.* $S_k = \{1, 3\}$. Then, this feature component must be added to $f_0$ when the reconstruction demand includes either the 1st or the 3rd image region. If the construction demand does not contains any region in $S_k$, then $\Delta f_k$ should not be added. Therefore, we can consider the selection of the feature component $\Delta f_k$ reflects an OR relationship between the target reconstruction regions.

Each feature component $\Delta f_k$ can be formulated as an OR interaction between reconstruction demands of different regions in $S_k$, as follows. Fortunately, we realize that we can extend the Harsanyi interaction theory (Ren et al., 2024a) to represent such OR interaction. It is because the OR relationship is equivalent to a specific AND relationship, *i.e.*, a feature component $\Delta f_k$ is **not** added if

---

**Algorithm 1** Disentangling feature components and find salient feature components

---

**Input:** Target image generated by a random input code $\mathbf{x} = g(d(\boldsymbol{z}))$, threshold $\tau$
**Output:** Feature components $\{\Delta f_i\}$, index set of salient feature components $\hat{\Omega}_N$
**for** $S \subseteq N$ **do**              ▷ computing minimal features
    compute minimal feature $u(S)$ based on Equation 9
**end for**
**for** $S \subseteq N$ **do**              ▷ computing OR interactions
    compute $\Delta f_k = I_{\text{or}}(S_k)$ based on Equation 10
**end for**
**for** $S_k \subseteq N$ **do**           ▷ selecting salient feature components using $\tau$
    **if** $\|\Delta f_k\|_{L-2} > \tau$ **then**
        The index set of salient feature components $\hat{\Omega}_N \leftarrow \hat{\Omega}_N \cup \{k\}, \Delta f_k = I_{\text{or}}(S_k)$
    **end if**
**end for**
return $\{\Delta f_i\}, \hat{\Omega}_N$

---

and only if **all** regions in $S_k$ are not in demand for reconstruction.

$$\Delta f_k = I_{\text{or}}(S_k) = -\sum_{S' \subseteq S_k} (-1)^{|S_k|-|S'|} u\left(N \backslash S'\right), S_k \neq \phi \tag{10}$$

Please see Algorithm 1 for the pseudo-code of disentangling feature components.

Just like the Harsanyi interaction (or the AND interaction), the above definition of the OR interaction satisfies the following universal property.

**Theorem 3.3.** *(Universal matching property of the OR interaction, proof in Appendix D). Given the demand of reconstructing regions in $\hat{S}$ in the target image $\mathbf{x}$, the minimal feature $u(\hat{S})$ for image reconstruction can be well estimated by a surrogate logical model $h(\mathbf{x}_{\hat{S}})$. The surrogate logical model sums up feature components corresponding to all interactions that are triggered by the reconstruction target $\hat{S}$.*

$$\forall \hat{S} \in N, u(\hat{S}) = h(\mathbf{x}_{\hat{S}}), \textit{ subject to}$$

$$h(\mathbf{x}_{\hat{S}}) \stackrel{def}{=} f_0 + \sum_{S_k \subseteq N, S_k \neq \emptyset} \mathbb{1}\left( \begin{matrix} \mathbf{x}_{\hat{S}} \textit{ triggers OR} \\ \textit{relation } S_k \end{matrix} \right) \cdot \Delta f_k = f_0 + \sum_{S_k \cap \hat{S} \neq \emptyset, S_k \neq \emptyset} \Delta f_k \tag{11}$$

Theorem 3.3 ensures that OR interactions for feature decomposition satisfy the Requirement 2, *i.e.,* when the demand of reconstructing regions in $\hat{S}$ will trigger all feature components $\Delta f_k$, as long as $\hat{S}$ partially covers any regions in $S_k$. Please see Algorithm 2 in Appendix E for a pseudocode of selecting a specific set $\hat{\Omega}_{\hat{S}}$ of feature components to reconstruct the target image regions in $\hat{S}$.

In addition, because the OR interaction can be considered as a special AND interaction that reverses the definition of the masked state and the unmasked state, the OR interaction also satisfies the sparsity property. The sparsity property shows that the minimal feature is the sum of a few non-zero feature dimensions in feature components, while other dimensions in feature components have negligible effects. The action region of a salient feature component can be view as a ***primitive regional pattern***. The entire image $\mathbf{x}$ is generated by superimposing these primitive regional patterns.

## 4 EXPERIMENT

We tested our method on the BigGAN-128 model (Brock, 2018). We used the intermediate features in the *layer2* for the feature decomposition, *i.e.,* the output of the first ResBlock in BigGAN. The baseline feature component $f_0$ was computed as the average feature over different input codes $\boldsymbol{z}$, *i.e.,* computing $f_0 = \mathbb{E}_{\boldsymbol{z}}[d(\boldsymbol{z})]$, when we set the DNN to generate images in different categories. Given an input code $\boldsymbol{z}$, the DNN generated the image $\mathbf{x}$, and the image was segmented into a $6 \times 6$ grid. Because the computation cost of interactions was NP-complete, we randomly selected $n = 9$ grids on the foreground object, and analyzed the OR interactions between these image regions encoded by the DNN.

## 4.1 Visualizing feature components (OR interactions)

Given a random input code $z$, we used the DNN to generate an image $\mathbf{x}$, and extracted feature components used for image generation. We followed the feature decomposition method described in Section 3. In order to obtain the baseline feature $f_0$, we used the DNN to generate different images in different categories, and extracted the average of intermediate features over all the generated images as the baseline feature $f_0$ in experiments. In order to compute the OR interaction corresponding each feature component, we divided the original images into $6 \times 6$ regions in grids, and randomly chose 9 regions of the main part of the image as input variables. The parameter $\alpha$ was initialized to a vector with all-ones elements, and $\lambda$ was set to 10000. We optimized $u(N)$, *i.e.*, the corresponding minimal feature of all regions in $N$. For each other set of regions $S \subseteq N$, the solution corresponding to $u(N)$ was set as the starting point of further computation of $u(S)$. Then we followed the algorithm in Equation 10 to compute $I_{\text{or}}(S_k)$. We used the L-2 norm $\|\Delta f\|_{L-2}$ to rank the interaction strength of each feature component (please see Appendix H for more detailed settings). Figure 5 shows the sparsity of the extracted interactions. This figure visualizes all elements of all the extracted interactions (feature components) by sorting their absolute values in a descending order. Most elements of feature components were almost zero, which verified the sparsity of the feature components.

Figure 3 verifies that the different feature components were exclusively responsible for the reconstruction of their own action regions. The interpretability of feature components enabled us to control the reconstruction of different image regions by adding different feature components.

## 4.2 Verifying the disentangled feature components (OR interactions)

**Towards the reconstruction of a single image region.** In order to verify whether the disentangled feature components were exclusively responsible for the reconstruction of specific image regions, we randomly selected a region $i \in N$ of the image $\mathbf{x}$. We chose all the feature components whose action regions included the $i$-th region. Then, we added these feature components into the baseline feature $f_0$, and obtained $\hat{f} = f_0 + \sum_{k \in \Omega} f_k, s.t. \, \Omega = \{k | i \in S_k\}$. Figure 7(a) shows image reconstruction results when we added the ratio $p$ of feature components that covered the $i$-th image region. We discovered that the feature $\hat{f}$ only well constructed the $i$-th region in the image $\mathbf{x}$. The target image region was gradually reconstructed when we added increasing numbers of feature components.

**Towards the reconstruction of a set of image regions.** We further evaluated our method in the reconstruction of a set of image regions. We randomly selected a set of regions $S \subseteq N$ in the image $\mathbf{x}$. We chose all the feature components whose action regions partially covered $S$. Then, we added these feature components into the baseline feature $f_0$, and obtained $\hat{f} = f_0 + \sum_{k \in \Omega} f_k, s.t. \, \Omega = \{k | S \cap S_k \neq \emptyset\}$. Figure 7(b) shows image reconstruction results when we added the ratio $p$ of feature components whose action regions partially covered $S$. We discovered that the feature $\hat{f}$ only well constructed the regions in $S$ in the image $\mathbf{x}$. The target image regions were gradually reconstructed when we added increasing numbers of feature components.

**Examining image reconstruction effects of irrelevant feature components**. From another perspective, we also examined whether irrelevant feature components would affect the generation of a region. We randomly selected the $i$-th region in image $\mathbf{x}$. Then, we gradually added all the feature components whose action region did not cover the $i$-th region, *i.e.*, $\hat{f} = f_0 + \sum_{k \in \Omega} \Delta f_k \, s.t. \, \Omega = \{k | i \notin S_k\}$. Figure 6 shows images reconstructed by adding the ratio $p$ of irrelevant feature components in $\Omega$. We found that although we had added all irrelevant feature components, the target image region was not changed.

## 5 Conclusion

In this paper, we introduce a new method to explain the internal representation of a image generation neural network. We use the OR interaction to disentangle components from the intermediate feature of the neural network. Our theory ensures that each component exclusively generates a primitive regional pattern, and the generation of the whole image can be explained as the superposition of all the extracted primitive regional patterns. Experiments have validated the faithfulness of the explanation

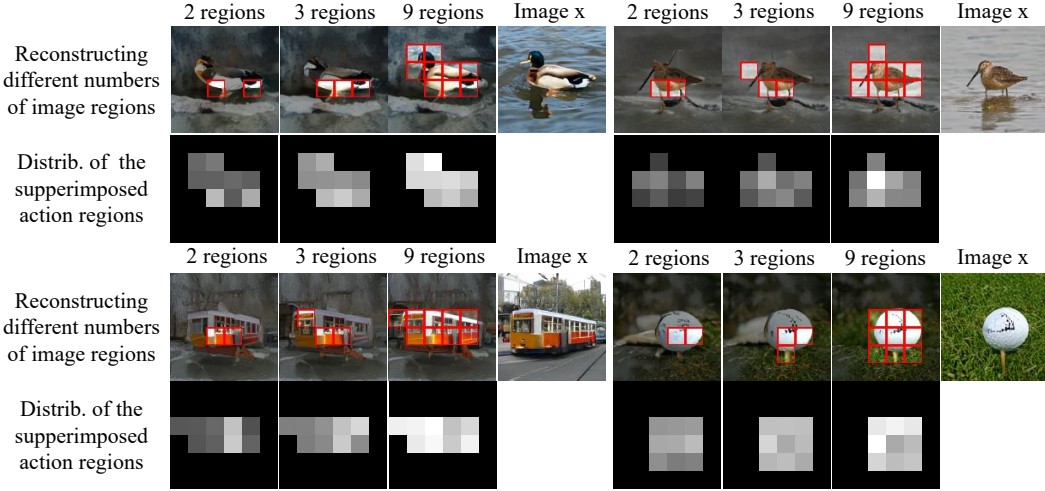

Figure 3: Incremental reconstruction of different image regions when we gradually added feature components. We added all feature components corresponding to the target image regions (in red boxes) for image reconstruction. It shows that different regions in the target object were sequentially reconstructed, but these feature components did not reconstruct the background. The heatmap shows the distribution of the overlapping action regions of the added feature components.

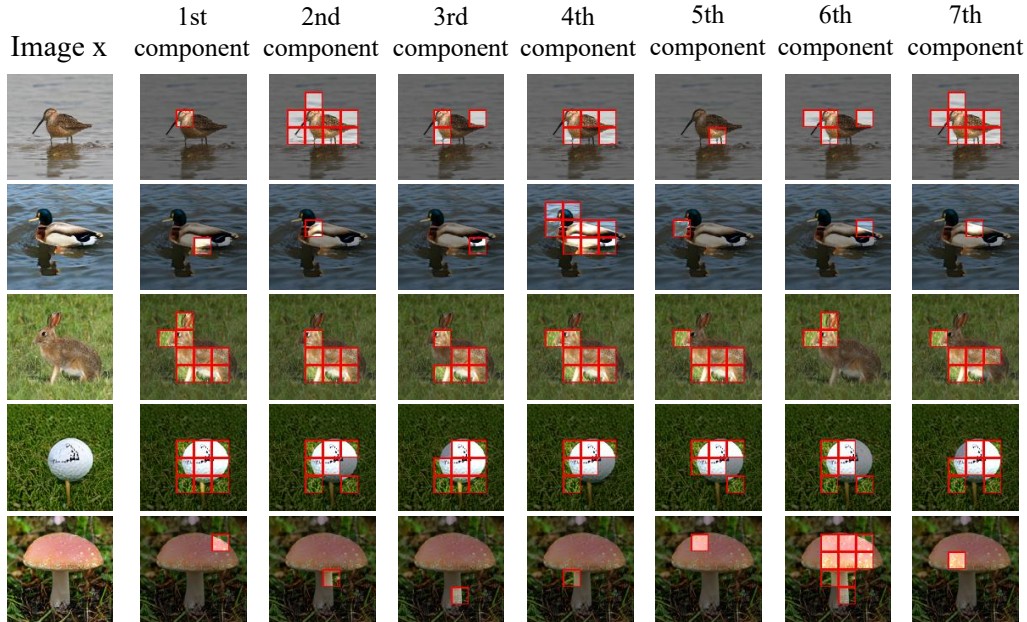

Figure 4: Action regions of the seven most salient feature components.

of the representation structure of the neural network. *I.e.*, each component is only responsible for the generation of a set of regions. People can control the neural network to exclusively reconstruct a specific set of image regions by adding feature components corresponding to these regions. As a limitation of the current theory, there is no strictly theory to constrain all feature components within the manifold of the intermediate-layer features that are generated by the input code $z$, although experiments have verified the effectiveness of using these feature components to control image generation. Thus, ensuring and computing the input code that corresponds to each combination of these feature components are the future work of this study.

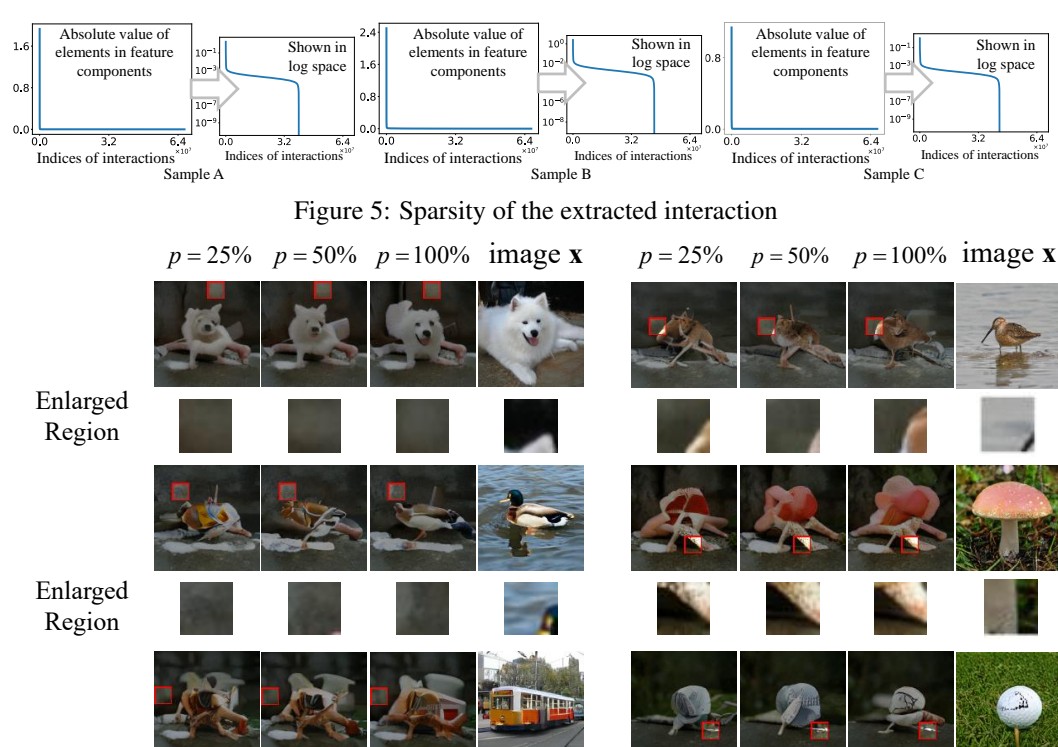

Figure 5: Sparsity of the extracted interaction

Figure 6: Validation of Requirement 1, *i.e.*, irrelevant feature components do not reconstruct the target region. We added different ratios $p$ of irrelevant feature components whose action regions did not cover the target image regions (in red boxes). We found that the target image region was not reconstructed, when we added these irrelevant feature components. Very weak reconstruction effects are caused by small computational errors.

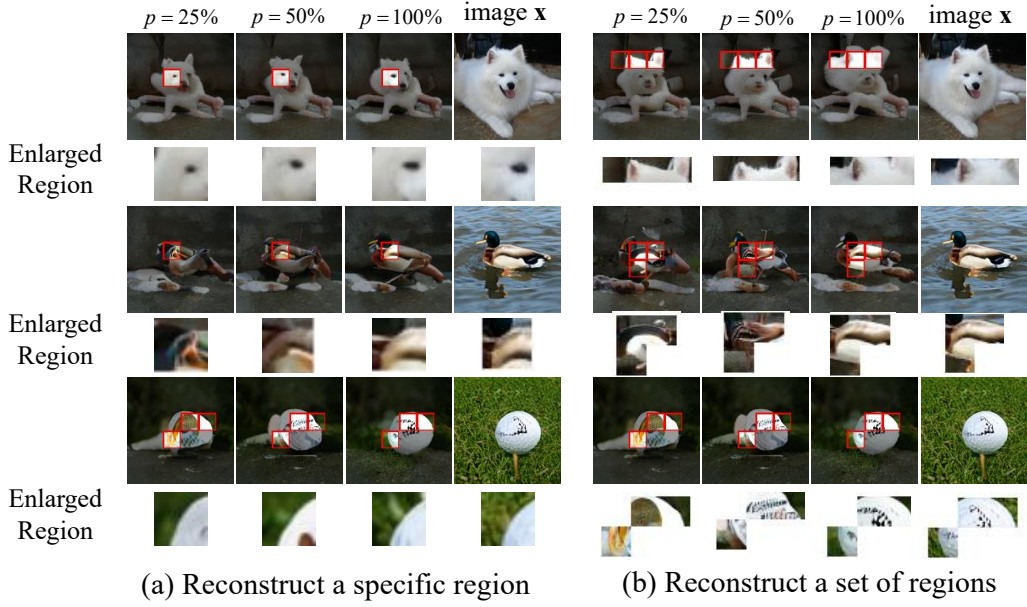

(a) Reconstruct a specific region     (b) Reconstruct a set of regions

Figure 7: Validation of Requirement 2, *i.e.*, relevant feature components reconstruct the target region. We added different ratios $p$ of feature components whose action regions covered the target image regions (in red boxes). We found that the target image region was gradually reconstructed when we added increasing number of feature components.

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

## A  AXIOMS AND THEOREMS FOR THE HARSANYI DIVIDEND INTERACTION

The Harsanyi dividend was designed as a standard metric to measure interactions between input variables encoded by the network. In this section, we present several desirable axioms and theorems that the Harsanyi dividend interaction $I(S)$ satisfies. This further demonstrates the trustworthiness of the Harsanyi dividend interaction.

The Harsanyi dividend interactions $I(S)$ satisfies the *efficiency, linearity, dummy, symmetry, anonymity, recursive* and *interaction distribution* axioms, as follows. We follow the notation in the main paper to let $u(S) = v(\mathbf{x}_S) - v(\mathbf{x}_\emptyset)$.

• **Efficiency axiom.** The output score of a model can be decomposed into interaction effects of different patterns, i.e., $u(N) = \sum_{S \subseteq N} I(S)$.

• **Linearity axiom.** If we merge output scores of two models $u_1$ and $u_2$ as the output of model $u$, i.e. $\forall S \subseteq N, u(S) = u_1(S) + u_2(S)$, then their interaction effects $I_{u_1}(S)$ and $I_{u_2}(S)$ can also be merged as $\forall S \subseteq N, I(S) = I_{u_1}(S) + I_{u_2}(S)$.

• **Dummy axiom.** If a variable $i \in N$ is a dummy variable, i.e., $\forall S \subseteq N \setminus \{i\}, u(S \cup \{i\}) = u(S)$ then it has no interaction with other variables, $\forall \emptyset \neq S \subseteq N \setminus \{i\}, I(S \cup \{i\}) = 0$.

• **Symmetry axiom.** If input variables $i, j \in N$ cooperate with other variables in the same way, i.e., $\forall S \subseteq N \setminus \{i, j\}, u(S \cup \{i\}) = u(S \cup \{j\})$, then they have same interaction effects with other variables, $\forall S \subseteq N \setminus \{i, j\}, I(S \cup \{i\}) = I(S \cup \{j\})$.

• **Anonymity axiom.** For any permutations $\pi$ on $N$, we have $\forall S \subseteq N, I_u(S) = I_{\pi u}(\pi S)$ where $\pi S = \{\pi(i) | i \in S\}$ and the new model $\pi u$ is defined by $(\pi u)(\pi S) = u(S)$. This indicates that interaction effects are not changed by permutation.

• **Recursive axiom.** The interaction effects can be computed recursively. For $i \in N$ and $S \subseteq N \setminus \{i\}$, the interaction effect of the pattern $S \cup \{i\}$ is equal to the interaction effect of $S$ with the presence of $i$ minus the interaction effect of $S$ with the absence of $i$, i.e., $\forall S \subseteq N \setminus \{i\}, I(S \cup \{i\}) = I(S|i\text{ is always present}) - I(S).I(S|i\text{ is always present})$ denotes the interaction effect when the variable $i$ is always present as a constant context, i.e. $I(S|i\text{ is always present})$ $= \sum_{S \subseteq I(S)} (-1)^{|S|} \cdot u(L \cup \{i\})$.

• **Interaction distribution axiom.** This axiom characterizes how interactions are distributed for "interaction functions". An interaction function $u_T$ parameterized by a subset of variables $T$ is defined as follows. $\forall S \subseteq N$, if $T \subseteq S, u_T(S) = c$; otherwise, $u_T(S) = 0$. The function $u_T$ models pure interaction among the variables in $T$, because only if all variables in $T$ are present, the output value will be increased by $c$. The interactions encoded in the function $u_T$ satisfies $I(T) = c$, and $\forall S \neq T, I(S) = 0$.

## B  PROVE FOR AND-INTERACTION UNIVERSAL MATCHING PROPERTY

In this section, we proof the univeral matching properties for the Harsanyi interaction.

**Theorem B.1.** *(Universal-matching property).Given an input sample $\mathbf{x}$, the output classification confidence $u(T)$ on each masked sample $\{\mathbf{x}_T \mid T \subseteq N\}$ can be well matched by a surrogate logical model $h(\mathbf{x}_T)$. The surrogate logical model sums up effects of all interactions that are triggered by the masked sample $\mathbf{x}_T$ as the output score.*

$$\forall T \subseteq N, h(\mathbf{x}_T) = u(T)$$
$$= \sum_{S \subseteq N, S \neq \emptyset} \mathbb{1}(\mathbf{x}_T \text{ triggers AND relation } S) \cdot I(S) \tag{12}$$
$$= \sum_{S \subseteq T, S \neq \emptyset} I(S)$$

*proof* According to the definition of the Harsanyi interaction, we have $\forall S \subseteq N$,

$$\sum_{T \subseteq S} I(T) = \sum_{T \subseteq S} \sum_{L \subseteq T} (-1)^{|T|-|L|} u(L)$$

$$= \sum_{L \subseteq S} \sum_{T \subseteq S: T \supseteq L} (-1)^{|T|-|L|} u(L)$$

$$= \sum_{L \subseteq S} \sum_{t=|L|}^{|S|} \sum_{\substack{T \subseteq S: S \supseteq L \\ |T|=t}} (-1)^{t-|L|} u(L)$$

$$= \sum_{L \subseteq S} u(L) \sum_{m=0}^{|S|-|L|} \binom{|S|-|L|}{m} (-1)^m$$

$$= u(S)$$

Therefore, we have $u(S) = \sum_{T \subseteq S} I(T)$.

## C PROVING THAT THE OR INTERACTIONS CAN BE CONSIDERED AS A SPECIFIC AND INTERACTION

The OR-interaction is defined as follow.

$$\Delta f_k = I_{\text{or}}(S_k) = -\sum_{S_l \subseteq S_k} (-1)^{|S_k|-|S_l|} u(N \backslash S_l), S_k \neq \phi \tag{13}$$

Here, $u(N \backslash S_l)$ denotes the minimal feature of the set $N \backslash S_l$, and regions in the set $S_l$ is not in the action region. Let us denote $\mathbf{x}_T$ as a masked state of $\mathbf{x}$, where only the regions in $T$ are presented. Furthermore, we define $\mathbf{x}'_T$ as the masked state where regions in $T$ are removed. Therefore, the definition of OR interaction can be rewritten as follows:

$$I_{\text{or}}(S \mid \mathbf{x}) = -\sum_{T \subseteq S} (-1)^{|S|-|T|} v(\mathbf{x}_{N \backslash T}), \quad T \neq \phi$$

$$= -\sum_{T \subseteq S} (-1)^{|S|-|T|} v(\mathbf{x}'_T), \quad S \neq \emptyset \tag{14}$$

$$= -I'_{\text{and}}(S \mid \mathbf{x}'), \quad S \neq \emptyset$$

where $v(\mathbf{x}_{N \backslash T}) = u(N \backslash T)$. Therefore, we can consider the OR interaction as a specific AND interaction.

## D PROOF OF THEOREM 3.3 IN THE MAIN PAPER

According to Appendix C, we reconsider the definition of the masked board state $\mathbf{x}_T$ as $\mathbf{x}'_T$. $\mathbf{x}_T$ denotes the masked state where regions in the set $T$ are inside the action region, and regions in the set $N \setminus T$ are not considered. In comparison, $\mathbf{x}'_T$ denotes the masked state where regions in the set $T$ are not considered, and regions in the set $N \setminus T$ are inside the action region.

In this way, the effect of an OR interaction based on the definition of $\mathbf{x}$ can be represented as the effect $I_{\text{or}}(S|\mathbf{x})$ of an AND interaction based on the definition of $\mathbf{x}'$.

$$I_{\text{or}}(S|\mathbf{x}) = w_S^{\text{or}} \cdot \left[ -\prod_{i \in S} \neg exist(x_i) \right]$$

$$= -w_S^{\text{or}} \cdot \prod_{i \in S} \neg exist(x_i) \tag{15}$$

$$= -\frac{w_S^{\text{or}}}{w_S^{\text{and}}} \cdot I'_{\text{and}}(S|x')$$

where the function $exist(x_i)$ represents whether $x_i$ is in the action region. Therefore, as we have proven that the universal-matching property of the AND-interaction, the OR interaction still satisfies this property.

---

**Algorithm 2** Reconstructing the regions in $\hat{S}$ based on all the salient feature components in $\hat{\Omega}_N$

---

**Input:** Target image regions $\hat{S}$, all the salient feature components $\hat{\Omega}_N = \{\Delta f_1, \Delta f_2, ...\}$
**Output:** Reconstructed image $\mathbf{x}_S$
$\hat{f} \leftarrow f_0$
**for** $k \in \hat{\Omega}_N$ do **do**
    **if** $S_k \cap \hat{S} \neq \emptyset$ **then**
        $\hat{f} \leftarrow \hat{f} + \Delta f_k$
    **end if**
**end for**
return $\mathbf{x}_{\hat{S}} = g(\hat{f})$

---

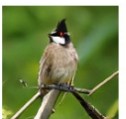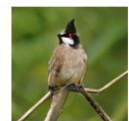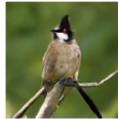

Figure 8: Images generated by same intermediate feature but different input codes

# E  A PSEUDOCODE OF RECONSTRUCTING IMAGE REGIONS

Algorithm 2 presents a pseudocode of selecting a specific set $\hat{\Omega}_{\hat{S}}$ of feature components to reconstruct the target image regions in $\hat{S}$.

# F  DISCUSSION FOR THE INFLUENCE OF THE INPUT CODE

The BigGAN model uses ResNet GAN architecture, *i.e.,* uses residual blocks, and the input code (a concatenated vector of the latent vector and the class vector) will be fed into the following modules. Therefore, after modifying the intermediate feature of a specific layer, the generated image might still change if the input code changed, due to the residual structure.

Therefore, to validate the influence of the input code is small, we randomly selected input codes, and modify the intermediate feature to a same $f$. The results are shown in Figure 8.

The experiment result showed that the influence of the input code is relatively small (the background might change a little, but the main parts of the image didn't change). Therefore, we can consider that most of the information is contained in the intermediate feature.

# G  THE IMPLEMENTATION DETAIL OF EXTRACTING FAITHFUL FEATURE COMPONENTS

Although we proved the sparsity of the OR interaction in the main paper, it is still challenging to regard those extracted regional patterns as the faithful primitive regional patterns encoded by the model. We find that the OR interaction is quite sensitive to the minimal feature and a small change to the minimal feature might not have a significant change in the generated image, but it will cause a change to the feature components we extracted.

Therefore, we ensured the interaction stability by removing the noise in the minimal feature. We use a vector $\boldsymbol{q}_S$ to model the noise in the minimal feature $\Delta f_k = I_o r(S)$. Each element in $\boldsymbol{q}_S$ is bounded in the range of $[-\tau, \tau]$, where $\tau$ is a threshold to avoid large noise.

Primitive regional patterns with stability is extracted by solving the following optimization problem:

$$\min_{\{\boldsymbol{q}_S\}} \sum_{S \subseteq N, S \neq \emptyset} \|I_{or}(S|\mathbf{x}, \{\boldsymbol{q}_S\})\|_{L-1} \tag{16}$$

In this way, we can ensure the interaction stability by removing noises in the minimal feature. Experiments in the main paper showed we can still well reconstruct the image after removing noises.

## H EXPERIMENT SETTINGS

This section includes detailed experiment settings in this paper.

In our experiment, we use BigGAN-128 to generate the image categories of ImageNet. Therefore, when calculating the baseline intermediate feature, we randomly generated images of all the 1000 classes in ImageNet. And the truncation of the BigGAN model was set to 0.4. After disentangling feature components, we ranked those feature components by their L-2 norm. In order to find out which components are salient, we removed the feature components from the intermediate feature in descending order, utill the generated image is similar to the baseline image. Empirically, we found that for most samples, the top 60% feature components were salient. And in Figure 5 we also found that in log space, about 40% interactions had relatively low interaction strength. When calculating the distribution of action regions, we performed a weighted sum of the number of feature components contained in each region, where the weight of a single feature component in a region is its L-1 norm divided by the number of its action regions. When we create heat map visualizations, we apply an exponential operation based on the weights mentioned above to enhance the contrast between different regions.

