# OpenReview forum: "Disentangling Regional Primitives for Image Generation"
_ICLR.cc/2025/Conference — ICLR 2025 Conference Withdrawn Submission_

### Official Review · Reviewer_e7ir · 2024-10-24

**Soundness:** 3
**Presentation:** 1
**Contribution:** 2
**Rating:** 3
**Confidence:** 3

**Summary:**

The paper presents an approach for extracting disentangled components that correspond to specific image regions. The paper provides a method that ensures that each such component is a minimal component that exclusively generates a specific set of image regions. Thus, the paper shows that the different components constitute an OR relationship for generating different parts of the image.

**Strengths:**

The paper presents an interesting approach for extending Harsanyi interaction for the image generative case. It provides a theory and builds an algorithm for matching the individual regions to disentangled representations. It finds and proves an interesting OR relationship that is implemented by the model via the disentangled features. The qualitative results show a clear correspondence between image regions and extracted features. This is a very interesting complementary result to the "AND" interaction (Ren et al., 2024a) that is implemented in discriminative models.

**Weaknesses:**

1. Presentation - My main concern is the clarity of the paper, which needs significantly more work. The paper is full of details and notations and is hard to follow. The presentation of the AND interaction is not directly related to this paper (e.g., Theorem 3.1/Corollary 3.2 are not mentioned anywhere after the preliminaries) and introduces more notation that is reintroduced later anyway (e.g., defining N twice). On the other hand, some of the notations are not defined (e.g., lambda in line 294). The introduction includes a lot of related work, and in general, there are a lot of repetitions in the text (e.g., between 4 and 4.1.). Given the current state of the paper, I couldn't verify all the proofs due to the difficulty of following the text. I recommend the authors significantly revise the paper to simplify a lot of the notations and remove redundant text.

2. Usefulness - Given that the extraction procedure is computationally expensive (worked on 6x6 patches, but what about larger images), it is not clear how practical this method is for extracting the disentangled representations. An analysis of the computational cost, the number of images, and other parameters that can make this approach practical is needed.

3. Interpretability - it is not clear that the disentangled representations are more interpretable to humans in any way. Although they are "minimal" features for regional generation, it does not mean that they are more interpretable -- they still have a complex functionality. Applying the same approach for being minimal about human-interpretable concepts (e.g., disentangling the minimal features that are responsible for foreground/background generation) will provide better interpretability and controllability to the generation process.

**Questions:**

1. Computation time - as mentioned, an analysis of the computational time about the number of regions/number of images is useful here and can tell how practical this approach is for finding disentangled directions.

2. Empirical analysis of the extracted components across a dataset - Do the extracted components create a pattern in the image space (e.g., some components are responsible for multiple regions while others are responsible for fewer regions)? How many components are needed to get different levels of disentanglement and fewer intersections between the corresponding regions (a graph that ablates it will be useful here)? Do more components correspond to the center of the image rather than the background?

---

### Official Review · Reviewer_1p7c · 2024-10-26

**Soundness:** 2
**Presentation:** 3
**Contribution:** 2
**Rating:** 5
**Confidence:** 4

**Summary:**

The paper introduces a new approach to explaining the internal workings of neural networks for image generation. The method breaks down intermediate-layer features into distinct components that correspond to specific image regions, aiming to interpret the model's internal representation as a combination of "primitive" regional patterns. By extending Harsanyi interaction theory to handle OR relationships between regions, the authors provide a novel way to link feature components with specific areas of the generated image. The experiments primarily use visual results from the BigGAN model to demonstrate the method's validity.

The paper does present a promising idea, but issues with the evaluation overshadow the strengths in problem formulation and theoretical discussion. Without more rigorous quantitative analysis, the contributions remain less convincing.

**Strengths:**

Fresh and Interesting Idea: The paper brings a new approach to explainable AI by breaking down image generation into basic regional patterns using OR interactions.

Strong Theoretical Foundation: The use of Harsanyi interaction theory is thoughtful and well-developed. It’s clear the authors put serious effort into the math, giving the approach a solid backbone.

Promising Results: While the evaluation could be better, the visual experiments do show the method can isolate specific image regions effectively. There’s a real potential here for fine-tuning or controlling generative models more precisely.

**Weaknesses:**

Despite I really like the first half of the paper, I would be somehow sceptical of the overall contribution for the following reasons.

1. Limited Quantitative Evaluation: ALL evaluations are visualisations. Quantitative metrics is indeed needed, like visual quality measurement, disentanglement testing. As it stands, we’re left guessing about the true effectiveness beyond what's visually apparent. Also, the absence of comparisons with other explainable methhods weakens the case for this approach.

2. Coarse Image Division: Dividing the images into a 6x6 grid feels too crude for explainability. There’s likely a lot of information lost with such a brutish approach. Why not use finer subdivisions or superpixels that adapt to the image content? That could make the whole decomposition more meaningful.

3. Unclear Visualization: The current visualizations fall short because the gray map overlays fail to highlight background changes, making it hard to see which regions are kept. Transparent overlays, heatmaps, or color-coding could make the results much clearer.

**Questions:**

Is the method applicable to other architectures, or does it depend heavily on the specific properties of BigGAN?

---

### Official Review · Reviewer_LhUX · 2024-11-04

**Soundness:** 2
**Presentation:** 1
**Contribution:** 2
**Rating:** 5
**Confidence:** 3

**Summary:**

The paper aims to provide an explanation framework for interpretation of the image synthesis process. The approach focuses on decomposing intermediate-layer features into "feature components," each responsible for generating specific image regions. By isolating these components, the method enables a new level of interpretability, where the image can be viewed as the superposition of "primitive regional patterns," each linked to distinct feature components.

**Strengths:**

* The paper analyzes the primitive component of the internal representation of a Deep Generation Network.
* The research falls in-line with the work of explaining the generation process of a distributed neural network via the idea of AND-OR graph [1] which is exciting. It contributes to the foundation of understanding the image generation process as hierarchical compositional theory.
* To model the OR interaction, the paper proposed to model it via Harsanyi's interaction theory. The paper provides formal writing about the theory that backs up the proposed algorithm.

[1] Xing, X., Wu, T., Zhu, S.C., Wu, Y.N., 2020. Inducing hierarchical compositional model by sparsifying generator networks. CVPR 2020.

**Weaknesses:**

* The author shows primarily the visual explanation of the generated image, however, it’s unclear how each “component” interacts with another component and how the components together compose the entire image in a hierarchy. It would be better if the authors can clearify what does mean by interaction?
* The authors also claim that the primitives interactions are minimized, however, Figure 4 shows that these components are heavily overlapped. Further clarification on the experiment part would be helpful.
* When explaining the feedforward process of a decision making network, it can make people understand how the network derived the conclusion. Although understanding unconditional image synthesis is interesting, it becomes much more impactful if the author could apply the proposed technique on image generation tasks that requires more precise control, such as image editing or image generation based on textual input, etc.
* The presentation of Algorithm 1 depends on the understanding of Equation 9 and 10. It would become much clearer if the author can decouple Equation 9 / 10 and provide an algorithm by simply showing the essential operational building blocks.
* A more quantitative evaluation protocol is needed to rigorously evaluate the effectiveness of the proposed method.

**Questions:**

* What benefits would the current method add to the overall image generation task?
* On what tasks could the proposed algorithm provide significant improvement?
* I don't understand the motivation of choosing specific regioins of the image to perform the analysis in Figure 3?
* Could you explain the definition of the "action region" in Figure 3?

---

### Official Review · Reviewer_9Wxt · 2024-11-04

**Soundness:** 3
**Presentation:** 3
**Contribution:** 2
**Rating:** 5
**Confidence:** 3

**Summary:**

This paper proposes a method to interprete the GAN model by disentangling regisonal primitives for image generation. Following the similar AND definition proposed in previous methods for classification models, this paper introduces OR definition for generation models. Some analysis validates that it shares some properties in common with AND. The authors also propose an algorithm to find salient feature components for an image. Some experiments demonstrate the methods can extract sparse interaction of regions within an image and yield plausible reconstruction results using extracted primitives.

**Strengths:**

Strengths:

1. It seems that the OR definition introduced in this paper is dual to the AND definition in previous works, which successfully extends previous framework for generative models.

**Weaknesses:**

Weaknesses:

1. To be honest, I cannot digest many parts of the article. This includes, but is not limited to, the tutorial of introduction, the method, and the experiments. I encourage the authors to rephrase some mathmatical descriptions with intuitive explanations.
2. The term $(-1)^{|S|-|T|}$ in many equations appear strange to me, e.g., Eq. 1 and Eq. 10. Why should we make the score for $T$ negative when the difference of the number of regions in $S$ and $T$ is odd? Please give more details on this -- it is important to understand its connection to the core idea of the submission.
3. I am not sure whether the extracted regional primitives are also applicable to generate other images. Otherwise, what is the purpose of extracting primitives for a specific image? The authors are encouraged to discuss the use cases of the method more clearly. Please provide  specific examples.
4. It seems that the algorithm needs enumerate each subset for a given set, which may be inefficient in practice. The authors are encouraged to discuss the time efficiency and hardware requirements specifically.
5. Is the addition defined in Fig. 1(c) element-wise? What is the tensor shape of each $\Delta f$? Does it represent a patch of it has the shape of a full image feature map, i.e., $C\times H\times W$? From my understanding, $f_0$ should have the shape $C\times H\times W$ and each $\Delta f$​ should also have the same shape for element-wise summation. If so, why does the visualization only contain a patch?
6. There is almost no quantitative results. Since reconstruction is one target, at least there should be some numerical evaluation regarding this, e.g., PSNR, SSIM, LPIPS, etc.
7. There is no comparison with other methods that also focusing on interpreting GAN. Please compare to the SOTA methods.
8. Why do the authors choose layer2 of BigGAN for validation? How does this choice influence the results?  How the results might change if different layers were used
9. The writing of the beginning part of Sec. 4 is largely overlapping with that of Sec. 4.1. Overall, the presentation, in terms of clarity, organization, and distinction with previous works, can be enhanced a lot.

**Questions:**

Please see the Weakness section.

Specifically, W2, W3, W5, and W8.

Please provide an answer in the rebuttal.

---

### Note · Authors · 2024-11-21

I have read and agree with the venue's withdrawal policy on behalf of myself and my co-authors.